# Effect of Date (*Phoenix dactylifera* L.) Pits on the Shelf Life of Beef Burgers

**DOI:** 10.3390/foods9010102

**Published:** 2020-01-18

**Authors:** Estrella Sayas-Barberá, Ana María Martín-Sánchez, Sarra Cherif, Jamel Ben-Abda, José Ángel Pérez-Álvarez

**Affiliations:** 1IPOA Research Group (UMH-1 and REVIV-Generalitat Valenciana), Agro-Food Technology Department, Escuela Politécnica Superior de Orihuela, Miguel Hernández University, Ctra. Beniel Km 3.2, E-03312 Orihuela, Alicante, Spain; ana.martin@umh.es (A.M.M.-S.); ja.perez@umh.es (J.Á.P.-Á.); 2High Agronomic Institute of Chott-Mariem, University of Sousse, B.P 47, Chott Mériem 4042, Tunisia; cherif.sarra@gmail.com (S.C.); jbenabda@yahoo.fr (J.B.-A.)

**Keywords:** safety, shelf life, burger, date pit, date palm coproduct, lipid oxidation, natural preservative

## Abstract

A new ingredient from date palm coproducts (pits) was obtained and tested as a preservative in burgers. Different concentrations of date pit (0%, 1.5%, 3%, and 6%) were added to beef burgers, and its effect on the safety and quality was evaluated during 10 days of storage. The incorporation of date pit was able to improve the shelf life and the cooking properties of the burgers. The date pit stabilized the color, lipid oxidation, and microbial growth of the burgers during the storage time due to the antioxidant activity and the phytochemical content of the date pits. For the consumer panel, the color and the off-odors were improved, and the addition of 1.5% and 3% date pit in cooked burgers obtained similar scores. Based on the obtained results, the new ingredient from date pit may have potential preservative properties for enhancing stability during shelf life and could improve the composition of bioactive compounds (fiber and phenolic content) of beef burgers.

## 1. Introduction

Consumers show an increasing demand for healthier food products, desirably free of chemical additives. Also, many studies have established some association between meat consumption and the incidence of health disorders such as coronary heart diseases and some types of cancers [1,2]. However, despite the concern related to food and nutrition and the current trends towards processed foods [3], burgers are still purchased worldwide, especially by younger consumers. For that reason, the meat industry must offer to these consumers “clean label” processed products or “free from” artificial additives/ingredients meat products [4] without affecting the safety and quality of the final product. 

Nevertheless, due to the nutritional composition of muscle tissue, meat and meat products are highly susceptible to quality deterioration, limited by microbial spoilage and lipid oxidation, mainly in processed products. The technological procedures during processing increase the risk of deterioration, which causes changes in color, texture, flavor, and nutritional quality. In these products, hygienic measures should be applied from the first stages of the process to avoid increasing the risk of deterioration (changes in color, texture, flavor, and nutritional quality) [5]. Burgers, as meat preparations, will undergo a higher level of oxidation and microbial contamination. Antioxidants and antimicrobials can be added to slow the oxidation process and microbial spoilage while increasing the safety and sensory quality of food. However, in recent years there is a growing concern about the potential toxicity and safety related to synthetic preservatives such as sulfites [6,7] or synthetic antioxidants such as butylated hydroxytoluene (BHT) [8]. Because of this, more research is being done on the use of natural ingredients, antioxidants, and antimicrobials [7,9,10,11]. 

Phenolic compounds are believed to be the main compounds with antioxidant properties, and antimicrobial activity has also been attributed to them. They are found in all parts of fruits, vegetables, and seeds, and many extracts have been investigated to reduce lipid oxidation and/or microbial spoilage in meat products [11,12,13]. Other strategies explored have been the use of essential oils [14,15,16], fiber concentrates [17,18], and spices [12].

The date palm (*Phoenix dactylifera* L.) is a typical crop from dry and semi-arid regions, so its main producers are located in the Middle East and North Africa. The annual production of dates reached more than 7 million tons in 2014 [19]. Date seeds constitute between 10% and 15% of the date fruit’s weight depending on the variety [20] and they are the primary by-products from the date processing industry (date syrup, date confectionery, pitted dates) [8]. This coproduct is mainly wasted or used as an animal feed. In the nutritional composition of dates, seeds highlight their bioactive compound content so that they could have important applications in the pharmaceutical [21] and food industries [20,22]. This valorization would represent an economical benefit for the cultivation and industrialization of dates and favor the sustainability of the sector.

Date pits (seed) are shown to be a good source of compounds with antioxidant [23] and antibacterial properties [24]. According to Al-Farsi and Lee [25] p-hydroxybenzoic, protocatechuic, and m-coumaric are the primary phenolic acids; likewise, they are also considered a source of dietary fiber [26], mainly water-insoluble fiber [20]. Consequently, date seeds could potentially be used as a functional ingredient in food to increase the fiber content [27]. Considering that they possess both antioxidant and antimicrobial activities, they may also be regarded as useful for maintaining meat safety and quality, extending shelf life. Therefore, date seeds could be regarded as a new ingredient for food use. It would be good to evaluate the viability of date seeds as a food enhancing agent to eliminate synthetic preservatives. Amany et al. [9] used date pit extract to evaluate the shelf life in a model meat system, but there is a lack of studies on the integral use of date pits for the refrigerated storage in meat products.

The objective of this study was to test the effect of a new ingredient from date pits on the shelf life of beef burgers.

## 2. Materials and Methods 

### 2.1. Date pit Preparation 

The date (*Phoenix dactylifera* L.) pits were prepared to be added as a powdered food ingredient. Date (cv. Deglet Nour) fruit seeds (pits), at the Tamar stage (fully ripe stage), were obtained from the date processing industry (Tunisian). They were transported to the laboratories of Miguel Hernández University (Orihuela, Spain). The pits were washed in water to remove date flesh and dried at 50 °C under vacuum. The dried material was ground using a stainless steel hammermill (IKA MF 10 basic, Staufen, Germany). A final 0.21 mm sieve was used to obtain the date pit powder (DPP). A final drying step during 2 h was repeated to eliminate the moisture from the powder. The DPP was characterized (proximate composition, phytochemicals, physicochemical and technological properties, and antioxidant activity) and then it was vacuum packaged and kept frozen until it was used. 

### 2.2. Characterization of Date Pit

#### 2.2.1. Proximate Composition

Moisture, protein, fat, and ash content were determined in triplicate using AOAC’s official methods [28]. Total dietary fiber (TDF) and the insoluble fraction (IDF) were determined by the enzymatic–gravimetric AOAC method 991.43 [28]. The difference between the TDF and the IDF corresponded to the soluble dietary fiber (SDF). 

#### 2.2.2. Physicochemical Analysis

Water activity was measured at 25 °C in a Novasina Thermoconstanter humidity meter (TH-500, Axair Ltd., Pfaeffikon, Switzerland). The CIELAB color space (L*: lightness; a*: redness/greenness; b*: yellowness/blueness) was determined using a spectrophotometer (Minolta CM-2600, Osaka, Japón, illuminant D65, 10° observer, SCI mode, 11 mm aperture of the instrument for illumination and 8 mm for measurement). A spectrally pure glass (CR-A51, Minolta Co., Osaka, Japan) was put between the samples and the equipment. Nine replicates of each sample were taken [29].

#### 2.2.3. Antioxidant Activity and Phytochemicals 

For antioxidant activity (AA), a range of 1 mg/mL to 50 mg/mL was prepared with dimethyl sulfoxide (DMSO). They were mixed for 1 min with a vortex and put in an ultrasonic bath for 30 min. The homogenates were centrifuged at 3500 rpm for 3 min at 4 °C, and the final extracts were collected after paper filtration. 

The total phenolic content (TPC) was determined following the Folin-Ciocalteu method [30] and expressed as gallic acid equivalents (GAE) per 100 g of fresh matter. The total tannin content and the AA were also measured by several methods on the DMSO extracts [31]. The AA assays were: the 2,2′-diphenyl-1-picrylhydrazyl (DPPH) radical scavenging capacity test, the ferrous ion-chelating activity (FIC), the reducing power and thiobarbituric acid reactive substances (TBARS). 

#### 2.2.4. Technological Properties

The technological properties of DPP were also evaluated: water holding capacity (WHC), oil holding capacity (OHC), water absorption capacity (WAC), emulsifying (EA) activity, emulsion stability (ES). All measurements were done in triplicate, and the methodology used for the different analyses was the same as previously described by Martín-Sánchez et al. [31].

### 2.3. Manufacture of Beef Burger

Fresh beef rib was obtained from a local market (Orihuela, Alicante, Spain). A basic burger formulation was used (90% beef rib, 8.5% water (ice), and 1.5% salt). The beef was ground (Mainca, Barcelona, Spain) using a 5 mm plate, and mixed with the rest of ingredients. The mixture was divided into four batches and date pit powder (DPP) was added at different concentrations: control (0% DPP), 1.5% DPP, 3% DPP, and 6% DPP. The different batches were mixed again for 5 min, including the control. 

Burgers were formed using a conventional burger maker (65–75 g/patty, 1 cm thickness and 9 cm diameter). Sixty-four burgers were prepared in each of the two independent manufacturing processes. Finally, they were stored for 10 days at 4 ± 1 °C in the dark in plastic bags (bag of 4 patties) in a relative vacuum (75%). Samples from each treatment were taken at 0, 3, 6, and 10 days for analyses. 

### 2.4. Beef Burger Analysis 

#### 2.4.1. Proximate Composition

The chemical composition (moisture, protein, fat, ash, and fiber) was determined on uncooked and cooked burgers on day 0, and AOAC’s official methods [28] were followed. Also, moisture and fat content were evaluated every sampling day in both raw and cooked patties for the determination of cooking properties.

#### 2.4.2. Physicochemical Analysis 

The pH was determined with a pH-meter (Model 507, Crison Instruments S.A., Barcelona, Spain) equipped with a thermometer and a combined electrode for solids (Cat. No. 52, Crison Instruments S.A., Barcelona, Spain). Color was determined from the surface of uncooked and cooked samples. Nine replicates of each sample were taken [31].

#### 2.4.3. Pigment Oxidation (Metmyoglobin Percentage)

The metmyoglobin percentage (MMb) was used as an indicator of pigment oxidation. MMb was determined on uncooked and cooked samples as described by Bekhit et al. [32] using the formula:%MMb = {−2.51(A572/A525) + 0.777(A565/A525) + 0.8(A545/A525) + 0.8(A545/525) + 1.098} × 100.

Samples (5 g) were homogenized with 25 mL phosphate buffer (40 mM, pH 6.8) during 10 s with an Ultra-Turrax homogenizer. The mixture was maintained for 1 h at 4 °C, centrifuged (4500 *g*, 30 min, 4 °C) and filtered. The absorbance was measured at 572, 565, 545, and 525 nm.

#### 2.4.4. Cooking Properties

Three burgers from each formulation were cooked at 163 °C in an oven to a core of 71 °C [33]. After cooking, they were cooled to room temperature (25 °C ± 1). The fat and moisture retained were estimated (g), samples were weighed (g), and the diameter measured (cm) before and after cooking to calculate cooking yield, diameter reduction, and retention of fat and moisture:Cooking yield (%) = (cooked weight/uncooked weight) × 100
Diameter reduction (%) = {(uncooked burger diameter − cooked burger diameter)/(uncooked burger diameter)} × 100
Retention (%) = {(cooked weight × component weight in cooked sample)/(uncooked weight × component weight in uncooked sample)} × 100.

#### 2.4.5. Lipid Oxidation

The extent of lipid oxidation was determined by the 2-thiobarbituric acid method in both samples, uncooked and cooked. Samples (0.5 g) were mixed with 2.5 mL of 0.375% thiobarbituric acid, 15% trichloroacetic acid, and 0.25 N HCl stock solution. After vortexing, samples were heated for 10 min in a boiling water bath and, after cooling, were centrifuged and filtered. The absorbance was measured at 532 nm. TBARS values were calculated from a standard curve of malonaldehyde (MDA) and expressed as mg MDA/kg sample. 

#### 2.4.6. Total Phenolics

Aliquots of the uncooked and cooked samples (2 g) were weighed into a centrifuge tube and 6 mL of DMSO was added. The mixture was then homogenized in ice for 1 min with the Ultra-Turrax homogenizer (15,000 rpm) and sonicated for 30 min in an ultrasonic water bath (Selecta S.A., Barcelona, Spain) in ice-cold water. Then, the mixture was centrifuged (12,000 rpm, 10 min, 4 °C) and filtered before determining the total phenolic content (TPC) using the Folin-Ciocalteau assay [30]. Results were expressed in mg GAE. Control samples would be the baseline for comparison in this assay, since other ingredients such as proteins may interfere. 

#### 2.4.7. Microbiological Analysis

Mesophilic aerobic (MAB; 30 °C for 48 h) and psychrotrophic aerobic bacteria (PAB; 7 °C for 10 days) were determined on a 3M Petrifilm^®^ Aerobic Count Plate (3M España S.A., Madrid, Spain). Enterobacteriaceae were determined on a 3M Petrifilm^®^ Enterobacteriaceae Count Plate with Violet Red Bile Glucose Agar (VRBG; 3M España S.A., Madrid, Spain) incubated at 37 °C for 24 h. Yeast and molds were done on potato dextrose agar (PDA) added with Chloramphenicol (Oxoid), incubated at 28 °C for 5 days (yeasts were counted after 72 h). All analyses were performed in triplicate and results were expressed as log CFU/g.

#### 2.4.8. Sensory Analysis 

Two sensory analyses were carried out separately, a visual and off-odor sensory analysis for the uncooked burgers each sampling day (trained panel) and one for the cooked burgers only on day 1 (consumer panel). A trained panel of seven people was selected from staff members of the AgroFood Technology Department at Miguel Hernández University. The burgers were presented in transparent Ziploc bags, firstly an odor evaluation (1 = none; 2 = slight; 3 = small; 4 = moderate, and 5 = extreme off-odor) was carried out immediately after opening the bag [34], and after 5 min, a color evaluation was done (1 = light red color; 2 = moderate light red color; 3 = acceptable red color; 4 = red-brown color, and 5 = dark grayish-brown). In addition, assessors were asked for the presence or absence of ropy slime.

As for the cooked burgers, a consumer panel (50 untrained panelists) from 18 to 59 years was asked about the intensity of color, off-odors, off-flavors, cohesiveness, hardness, juiciness, particle detection (undetectable to highly detectable) on a scale from 0 to 7, and final global acceptability on a hedonic scale (1—Dislike extremely, 2—Dislike moderately, 3—Dislike slightly, 4—Neither like nor dislike, 5—Like slightly, 6—Like moderately, and 7—Like extremely highly). Rectangular pieces (2 cm ± 2 cm) of cooked burger (Section 2.4.4.) were cut and served at room temperature [34]. Crackers and mineral water were also provided. Both analyses were performed according to the specifications of the International Standards Organization [34] and the American Meat Science Association [33].

### 2.5. Statistical Analysis

Statistical analyses were carried out using the statistical package SPSS 20.0 (IBM SPSS Statistics 20, IBM Corporation, Somers, NY, USA). Conventional statistical methods were used to calculate means and standard deviations. The effect of the formulation (control, 1.5% DPP, 3% DPP, and 6% DPP) and storage time (days 0, 3, 6, and 10) was analyzed by a two-way analysis of variance (ANOVA). The Tukey post hoc test was used for comparison of means, and differences were considered significant at *p* < 0.05.

## 3. Results and Discussion

### 3.1. Characterization of Date Pit Powder (DPP)

Dietary fiber, fat, and phenolic compounds were the main components of DPP (Table 1). Most of the fiber (96%) corresponded to the insoluble fraction [20], and most of the fatty portion tended to be oleic acid [35,36,37]. The chemical composition was similar to that found in other date pit varieties [26,38]. The physicochemical properties obtained were comparable to other dried fibers from other by-products [37,39]. 

More than 9% of DPP corresponded to phenolic compounds, which agrees with the results in the acetone–water extracts obtained by Al-Farsi and Lee [25]. Tannins represented 37% of TPC; therefore, the presence of these compounds, together with the high AA shown by DPP (Table 1), justifies considering DPP as an antioxidant dietary fiber [38]. Also, DPP showed very high values of radical scavenging capacity, chelating activity, reducing power, and inhibition of lipid peroxidation, when compared with other co-products of the date [31].

DPP showed a considerable WHC (1.6 times its own weight) and a medium OHC (Table 1). The hydration property may help to provide a pleasant texture and reduce syneresis and dehydration during storage, while a high OHC would provide a non-greasy sensation, helping to stabilize fatty products [40]. However, DPP may be a good emulsifying agent, with an emulsion stability of 100%. According to Akasha et al. [40], the seed proteins are associated with polysaccharides (as glucans, xylans, cellulose, and mannans) and affect their functional properties, promoting their capacity as an emulsifier. The emulsifying activity is indicative of its ability to adsorb biliary acids, limiting their intestinal absorption and therefore reducing blood cholesterol. DPP presented a similar EA to that of chia fiber [39]. 

These results support the use of date pits as a novel functional ingredient for the food industry, and as an excellent source of dietary fiber and phenolic antioxidants [20,22,27,36,38].

### 3.2. Characteristics of Burgers with Date Pit Powder (DPP)

Date pits showed a direct effect on moisture, with a significant decrease proportional to DPP content due to the increase in dry matter (Table 2). The addition of 6% DPP reduced moisture to nearly 4% in the uncooked and cooked burgers. 

No statistical differences were found between treatments for the fat and ash content, while 6% of added DPP presented a smaller protein content (*p* < 0.05) than the control. Moisture loss during cooking affected mainly the protein content, increasing it by 5%, and this effect has also been observed in other studies [38]. In the cooked burger, the addition of 6% DPP reduced the moisture and protein content. This could be due to the presence of non-meat particles (6% DPP) that would decrease the binding in the meat matrix, while small percentages (1.5% DPP) strengthen the binding [17]. The fat did not suffer an increase after cooking due to the fat drip loss. Fiber content increased (*p* < 0.05) proportionally to the percentage of DPP included, and only the addition of 6% DPP would allow classifying this product as a source of fiber (at least 3 g of fiber per 100 g DPP). 

All uncooked burgers had similar pH values (Table 2). Cooking slightly increased pH values (from 5.5–5.6 until 5.98) in all samples; therefore, it was due to the effect of cooking on the meat cells. Yıldız-Turp and Serdaroglu [41] also observed this increase, attributed to the breaking down of the cellular buffer ability and free fat. 

### 3.3. Color Deterioration and Pigment Oxidation of Burgers with DPP During Storage

The addition of DPP and the effect of the storage time significantly affected the color of uncooked and cooked burgers (*p* < 0.05) (Table 3). The cooking treatment obviously changed color parameters, as a consequence of the Maillard reaction, protein denaturation, water and fat loss. 

Increasing the level of DPP decreased the L* values (*p* < 0.05), probably as a consequence of the lower moisture proportion when fiber was added, since moisture is related to lightness values [38,39,42]. During the refrigerated storage, the L* values in all cooked burgers decreased, probably due to changes in the meat matrix that would involve modifications in the free water [43]. Tannin oxidation may generate some darkening as well, which occurs more rapidly at elevated temperature [43]. On days 3, 6, and 10 the yellowness of samples with DPP presented higher values than the control when they were uncooked, but lower when they were cooked. Therefore, the cooking process affected yellowness, a parameter that has also been related to myoglobin changes [42].

A decrease in redness makes meat products less attractive to consumers. Control burgers underwent a strong discoloration during storage (*p* < 0.05), being visually noticeable as well. However, the addition of DPP had a protective effect on color during storage, with more intensity in samples with 6% DPP added. Other authors adding antioxidant extracts have also been able to minimize redness changes during storage [44]. 

The discoloration of ground meat has been attributed to myoglobin oxidation induced by pro-oxidant substances released during lipid oxidation [31], and to protein oxidation among other causes [44]. Thus, the metmyoglobin formed (brownish color) could be used as an indicator of pigment oxidation. The incorporation of DPP had a very intense protective effect against myoglobin oxidation (Table 3) in raw samples from day 0 (*p* < 0.05). The MMb formed in control samples reached 51.0%, whereas the formulas with DPP presented values ranging from 9.8% to 16.0%. Moreover, in raw burgers, the decrease in redness followed the same pattern as the increase in MMb. This same protective effect of DPP against color deterioration and pigment oxidation (decrease a* and MMB) has been observed in synthetic preservatives such as sulfite [45], which supports the hypothesis of the use of date pits as a natural preservative for beef burgers.

All cooked burgers presented MMb percentage in a proximate range, probably due to changes or interactions occurring in the meat product during cooking, which are related to pigment oxidation. If after cooking the burger showed standard red-brownish color, it could be due to the formation of denatured-globin hemochromes and Maillard products, the state of the protein and other components [46].

### 3.4. Cooking Parameters of Burgers with DPP During Storage

The incorporation of date pits in beef burgers as a functional ingredient affected positively the cooking properties, and this behavior depended on concentration (Table 4). Cooking parameters are fundamental in this type of meat product because these parameters affect the quality, nutritional value (loss of soluble vitamins and amino acids), texture and juiciness, and acceptance of the product [47].

Burgers formulated with 3% and 6% of DPP presented a higher cooking yield (*p* < 0.05), around 6% more. In general, during the first 6 days of storage, the presence of date pits at high concentrations (3% and 6%) in burgers favored the retention of fat and moisture; this could suggest the existence of a stronger meat matrix structure in the burger with these added concentrations [39]. But this effect could not be observed for the concentration of 1.5% DPP. To ensure the sensory quality and acceptability of a fresh meat product, it is necessary to maintain the fat within the matrix of the product during cooking and storage [48]. 

During cooking, meat products shrink (diameter reduction percentage) due to protein denaturation, evaporated water, and fat loss. These effects were similar to other studies with fiber-rich ingredients [38,47]. The addition of DPP reduced the diameter reduction in all cases in comparison with the control. These lower values could be related to a higher retention of water and fat in the meat matrix [49,50] caused by the date pit fiber. Positive results were also reported in burgers with other fiber [17,47]. Therefore, DPP could avoid the dramatic change, cooking loss, and diameter reduction in the burger, which contribute to a bad reaction from consumers.

### 3.5. Lipid Oxidation and Total Phenolic Content (TPC) of Burgers with DPP During Storage

The TBARS and the TPC results are shown in Figure 1. TPC was significantly higher (*p* < 0.05) in samples with DPP, and it was clearly proportional to added DPP content and the contents remained constant during storage (*p* > 0.05). Control burgers presented the highest values of lipid oxidation during refrigerated storage (*p* < 0.05), whereas formulas with 1.5% and 3% DPP were more efficient as antioxidants. Adding high concentrations of antioxidants can have pro-oxidant activity. Therefore, in this study, the reduction of lipid oxidation was not proportional to the content of phenols, so that other factors would be intervening in lipid oxidation. 

When plant phenolic compounds are added to a food system, their activity is influenced by chemical composition, environmental conditions, interactions with lipids, the presence of other active substances, the ability to reduce iron and scavenging radicals among others [51]. The incorporation of 6% DPP did not show pro-oxidant activity in comparison with the control, although their values were higher than the samples with 1.5% and 3% DPP. According to the results of this study, using concentrations below 6% to prevent oxidation will be more advantageous. This may explain why the MMb was lower in samples with 1.5% DPP since lipid oxidation would also be related to pigment oxidation. 

Many authors have also found an antioxidant effect on meat products from the different fibers, by-products, and phenolic-rich extracts [12,44]. Date pits are rich in tannins, and the effect of tannic acid on ground beef (200 mg/kg) has been previously studied by Maqsood and Benjakul [52]. They found very similar TBARS and MMb results for the control samples, and also found a protective effect against lipid and myoglobin oxidation. Therefore, phenolic compounds, including DPP tannins, were probably responsible for the antioxidant activity. Liquid extracts from date pits have also been tested on ground beef for their antioxidant activity [8].

### 3.6. Microbial Quality of Burgers with DPP during Storage

The PAB, MAB, and enterobacteria counts (Figure 2) increased in all samples during storage (*p* < 0.05), and all started with microbial counts lower than 5 log CFU/g, a common value for fresh meat products prepared by mincing and grinding. PAB was the most reduced microbial group (*p* < 0.05) by the presence of DPP. The use of 3% DPP was the most effective against PAB, with less than 5 log CFU/g during the 10 days of storage, nearly 2 log CFU/g less than the control. 

Adding 3% of DPP protected against MAB growth from day 6 (*p* < 0.05), while samples with 6% of DPP showed higher protection from day 3 (*p* < 0.05). Only control samples reached counts of 7 log CFU/g at the end of storage (10 days), indicating spoilage, which was also detected by the trained panel with the highest off-odors scores (Section 3.7). 

The 3% of DPP had a higher effect on enterobacteria, this concentration reached less microbial counts during storage. The counts of enterobacteria were higher for the 6% of DPP than in the control at the beginning of the storage, although both counts were similar at the end of storage.

Other studies have suggested that date pit polyphenols have a certain antibacterial activity in vitro, although there are no studies carried out on meat products. Date pit extracts have been evaluated for their antibacterial activity in vitro against different bacteria, showing positive results against Gram+ and Gram− bacterias (except *Enterococcus faecalis*) due to the flavonoids present in such extracts, which showed antimicrobial properties [24]. Maqsood and Benjakul [52] evaluated the effect of adding tannic acid to ground beef, one of the types of phenolics present in important amounts in date pits; and they also found that it was effective to slow microbial growth. Therefore, considering the results of the three microorganisms assessed, the use of DPP in a concentration of 3% would be more efficient to slow microbial growth, as it obtained lower counts than the control sample. The phenolic compounds may show inhibition or activation of microbial growth and metabolism according to their constitution and concentrations. 

### 3.7. Sensory Evaluation

For the visual color and the off-odors of the uncooked burgers (trained panel), control samples had the highest scores (*p* < 0.05) (more discolored/grayish and more undesirable odors) during storage (from 3-day to 10-day) (Table 5). On day 0 the color and off-odor differences were not highly significant between control and samples with DPP, but the results from day 6 showed a stabilizing effect of DPP on color and odor. The scores of the visual color on day 10 of storage were higher for control followed by 1.5%, 3%, and 6% DPP (Table 4); these results were in accordance with the findings reported for the instrumental color (redness) and MMb, indicating that addition of DPP protected against discoloration, and this was dependent on the concentration (Table 3). Control samples on day 10 showed the highest off-odors scores against DPP samples. The ropy slime was very similar in all formulas; it was not observed in the patties during storage.

Consumer panel results for cooked burgers at day 0 with different DPP levels are shown in Figure 3. The addition of DPP did not affect (*p* > 0.05) the aroma, flavor, particle detection, cohesiveness, and acceptability. Therefore, the different formulas did not show off-odors/flavors; DPP was not perceived in the mouth; and the cohesiveness, as the texture values indicated, was very similar in all formulas when the product was cooked. Regarding the overall acceptance, differences were not significant, it was very similar to nearly all treatments (around 5.5), but the 6% DPP burgers received the lowest score (4.5). Significant differences were found for the rest of the parameters evaluated. The visual color was perceived as darker in 6% DPP samples, it agrees with its lowest value of L* (Table 3). The rest of the burgers had a redder shade. Finally, the burgers with 6% DPP showed the lowest juiciness values and the highest hardness scores, which agrees with the lower moisture content (Table 2) and the water-holding capacity of date pit fiber, which contributed to hardness perception increased. 

## 4. Conclusions

Date pits could be considered as a potential non-meat ingredient in fresh meat products, improving their preservation and increasing their fiber content. In general, adding 3% of date pit powder to beef burgers would be the most advantageous concentration, since the phytochemical and fiber content of pits were able to preserve the red color of raw burgers, avoid pigment and lipid oxidation, reduce microbial counts, and improve the cooking properties. Also, burgers with 3% DPP led to the highest acceptability of the product to the trained panel, and after cooking they showed one of the best sensory profiles. Date pits could have significant potential as a food improvement agent to eliminate synthetic preservatives and to enrich foods with fiber without losing flavor, color, and texture.

## Figures and Tables

**Figure 1 foods-09-00102-f001:**
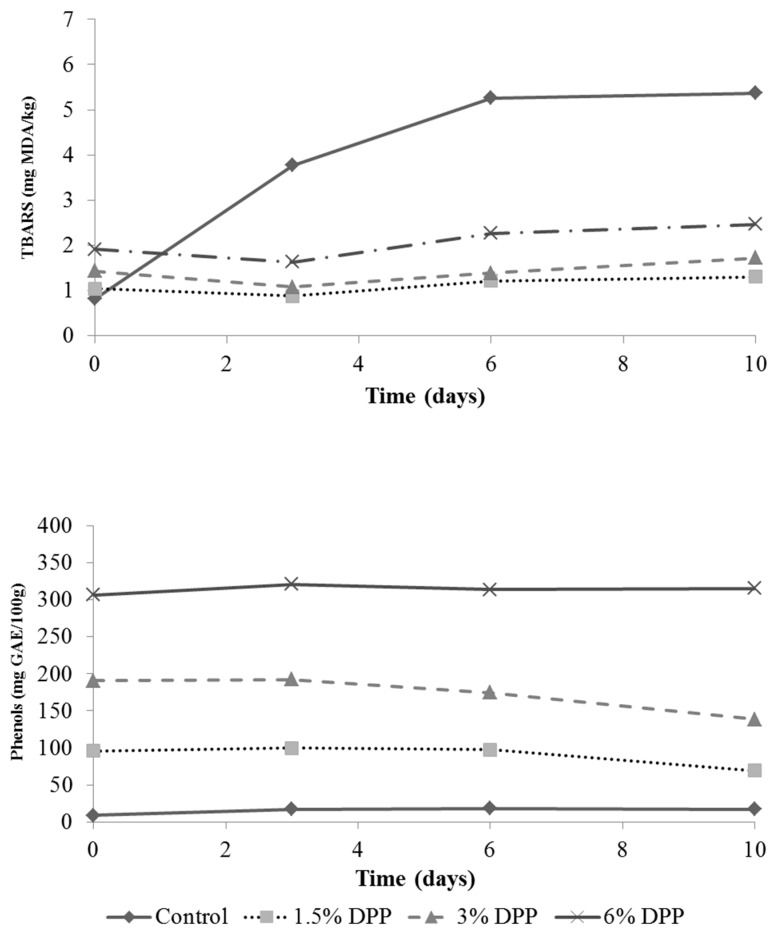
Evolution of the lipid oxidation (TBARS) and total phenolic content of uncooked burgers with date pit powder (DPP) during refrigerated storage. GAE: Gallic acid equivalents; TBARS: Thiobarbituric-acid-reactive substances; MDA: Malonaldehyde.

**Figure 2 foods-09-00102-f002:**
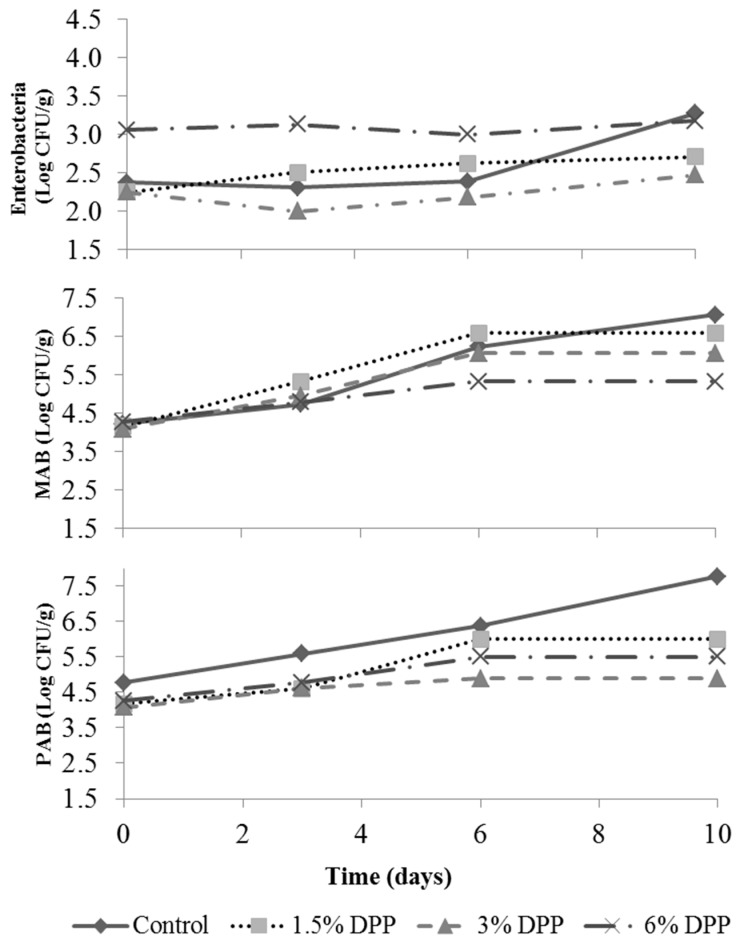
Evolution of the mesophilic aerobic bacteria (MAB), enterobacteria, and psychotropic aerobic bacteria (PAB) of the uncooked burgers with date pit powder (DPP) during refrigerated storage.

**Figure 3 foods-09-00102-f003:**
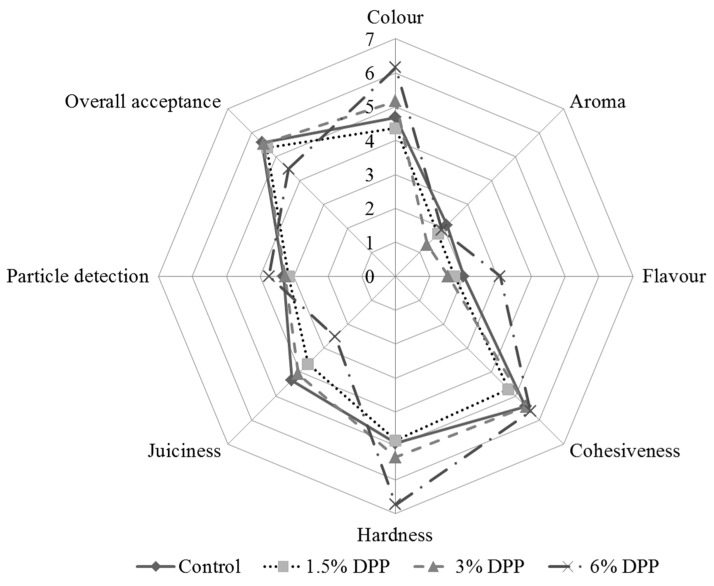
Results of the sensory analysis (consumer panel) in cooked burgers with date pit powder (DPP).

**Table 1 foods-09-00102-t001:** Characterization of date pit coproduct.

**Chemical Composition (%)**		**Technological Properties**	
Moisture	4.72 ± 0.24	WHC (g water/g DPP)	1.59 ± 0.02
Protein	4.77 ± 0.34	OHC (g oil/g DPP)	0.82 ± 0.12
Ash	0.89 ± 0.18	WAC (g water/g DPP)	0.42 ± 0.15
Fat	9.84 ± 0.02	EA (mL/100 mL emulsion)	50.7 ± 2.4
Total dietary fiber	70.11 ± 1.85	ES (%)	100
Insoluble dietary fiber	67.33 ± 2.01	**Phytochemicals and Antioxidant Activity**	
Soluble dietary fiber	2.78 ± 2.01	Total phenolic content (g GAE/100 g)	9.17 ± 0.04
**Physico** **chemical Properties**		Tannins (g GAE/100 g)	3.36 ± 0.10
a_w_	0.201 ± 0.003	DPPH (mM TE/100 g)	63.2 ± 9.6
L*	61.55 ± 0.45	FIC (µM EDTA/100 g)	78.3 ± 10.8
a*	9.72 ± 0.21	Reducing power (mM TE/100 g)	70.8 ± 0.9
b*	18.07 ± 0.12	TBARS (µM BHT/100 g)	14.5 ± 0.2

WHC: water holding capacity; OHC: oil holding capacity; WAC: water absorption capacity; EA: emulsifying activity; ES: emulsion stability; DPP: date pit powder; a_w_: water activity; L*: lightness; a*: redness; b*: yellowness; GAE: gallic acid equivalents; DPPH: 2,2’-diphenyl-1-picrylhydrazyl; TE; trolox equivalents; FIC: ferrous ion-chelating activity; EDTA: ácido etilendiaminotetraacético; TBARS: thiobarbituric acid-reactive-substances; BHT: butylated hydroxytoluene.

**Table 2 foods-09-00102-t002:** Chemical analysis and pH of burgers (uncooked and cooked) with date pit powder.

**Uncooked Burgers**	**Control**	**1.5% DPP**	**3% DPP**	**6% DPP**
Moisture (%)	73.8 ± 1.1 ^b^	72.7 ± 1.3 ^ab^	72.3 ± 0.5 ^ab^	70.3 ± 0.9 ^a^
Fat (%)	4.8 ± 0.4	5.6 ± 0.4	5.0 ± 0.3	4.9 ± 0.1
Protein (%)	19.6 ± 0.7 ^b^	18.7 ± 0.4 ^b^	18.6 ± 0.6 ^b^	17.0 ± 0.5 ^a^
Ash (%)	2.3 ± 0.2	2.2 ± 0.8	2.4 ± 0.7	2.2 ± 0.1
Fibre (%)	0.0 ± 0 ^a^	0.8 ± 0.1 ^b^	1.8 ± 0.1 ^c^	4.2 ± 0.2 ^d^
pH	5.50 ± 0.01	5.53 ± 0.01	5.52 ± 0.01	5.51 ± 0.01
**Cooked Burgers**	**Control**	**1.5% DPP**	**3% DPP**	**6% DPP**
Moisture (%)	66.0 ± 0.1 ^b^	67.3 ± 0.1 ^c^	65.8 ± 0.2 ^b^	63.1 ± 0.3 ^a^
Fat (%)	4.9 ± 0.4	5.0 ± 0.4	5.3 ± 0.4	5.0 ± 0.1
Protein (%)	25.7 ± 0.5 ^c^	23.7 ± 0.1 ^b^	23.0 ± 0.1 ^b^	22.3 ± 0.5 ^a^
Ash (%)	2.5 ± 0.1	2.5 ± 0.3	2.4 ± 0.7	2.4 ± 0.5
Fibre (%)	0.0 ± 0.0 ^a^	1.0 ± 0.1 ^b^	1.9 ± 0.1 ^c^	5.5 ± 0.1 ^d^
pH	5.90 ± 0.01	5.93 ± 0.02	5.94 ± 0.01	5.92 ± 0.01

Mean in same row with different letters are significantly different between percentages of added DPP (*p* < 0.05); DPP: date pit powder.

**Table 3 foods-09-00102-t003:** Color properties and metmyoglobin percentage (MMb) of burgers (uncooked and cooked) with date pit powder (DPP) during refrigerated storage.

**Uncooked Burgers**	**Day**	**Control**	**1.5% DPP**	**3% DPP**	**6% DPP**
**L***	0	^B^ 53.3 ± 2.5 ^b^	^C^ 50.2 ± 2.1 ^a^	^B^ 52.1 ± 1.3 ^ab^	^B^ 53.8 ± 1.9 ^b^
	3	^A^ 41.6 ± 1.2 ^a^	^AB^ 43.4 ± 2.8 ^ab^	^A^ 44.2 ± 3.3 ^ab^	^A^ 45.5 ± 1.5 ^b^
	6	^A^ 44.2 ± 3.3 ^b^	^A^ 41.2 ± 1.6 ^a^	^A^ 43.9 ± 1.6 ^ab^	^A^ 43.1 ± 1.2 ^ab^
	10	^A^ 43.9 ± 1.1	^B^ 44.9 ± 6.3	^A^ 43.9 ± 1.0	^A^ 45.9 ± 4.4
**a***	0	^B^ 14.00 ± 3.25 ^b^	^B^ 14.46 ± 1.31 ^b^	^B^ 14.09 ± 0.82 ^b^	^B^ 12.65 ± 1.07 ^a^
	3	^A^ 7.26 ± 0.80 ^a^	^B^ 9.17 ± 1.24 ^b^	^A^ 9.88 ± 1.76 ^b^	^A^ 10.19 ± 0.64 ^b^
	6	^A^ 5.01 ± 0.84 ^a^	^B^ 10.95 ± 0.94 ^b^	^A^ 9.94 ± 0.86 ^b^	^A^ 10.43 ± 0.49 ^b^
	10	^A^ 3.97 ± 0.36 ^a^	^A^ 5.11 ± 0.79 ^a^	^A^ 8.67 ± 0.60 ^b^	^A^ 9.84 ± 1.97 ^b^
**b***	0	^B^ 15.7 ± 2.8	^C^ 13.7 ± 0.9	^B^ 13.7 ± 0.6	^B^ 13.6 ± 1.2
	3	^A^ 8.8 ± 0.7 ^a^	^B^ 10.5±1.0 ^b^	^A^ 10.3 ± 2.1 ^ab^	^A^ 11.3 ± 0.9 ^b^
	6	^A^ 8.9 ± 2.0 ^a^	^B^ 11.7 ± 0.9 ^bc^	^A^ 10.3 ± 0.9 ^ab^	^AB^ 12.4 ± 1.5 ^c^
	10	^A^ 8.3 ± 0.7 ^a^	^A^ 7.9 ± 1.2 ^a^	^A^ 9.8 ± 0.7 ^b^	^AB^ 12.6 ± 1.2 ^c^
**MMb**	0	^A^ 51.02 ± 2.69 ^b^	^A^ 16.03 ± 1.74 ^a^	^A^ 12.93 ± 1.69 ^a^	^A^ 9.77 ± 0.04 ^a^
	3	^B^ 67.14 ± 1.08 ^b^	^A^ 22.05 ± 2.28 ^a^	^B^ 20.20 ± 2.79 ^b^	^B^ 17.26 ± 1.34 ^b^
	6	^B^ 73.24 ± 4.79 ^b^	^A^ 21.59 ± 0.99 ^a^	^B^ 18.11 ± 0.17 ^b^	^B^ 16.95 ± 0.70 ^b^
	10	^B^ 71.82 ± 0.67 ^b^	^B^ 62.50 ± 1.23 ^b^	^B^ 24.58 ± 3.32 ^b^	^B^ 19.87 ± 0.86 ^b^
**Cooked burgers**	**Day**	**Control**	**1.5%DPP**	**3%DPP**	**6%DPP**
**L***	0	^D^ 64.6 ± 0.2 ^c^	^C^ 62.9 ± 0.5 ^b^	^C^ 61.7 ± 0.4 ^a^	^C^ 61.3 ± 0.5 ^a^
	3	^B^ 55.1 ± 0.4 ^c^	^A^ 51.6 ± 0.2 ^a^	^A^ 52.2 ± 0.2 ^a^	^B^ 52.2 ± 0.9 ^ab^
	6	^A^ 53.9 ± 0.6 ^c^	^B^ 52.9 ± 0.4 ^ab^	^B^ 53.0 ± 0.3 ^b^	^B^ 52.3 ± 0.4 ^a^
	10	^C^ 56.1 ± 0.3 ^d^	^B^ 52.6 ± 0.4 ^c^	^A^ 52.1 ± 0.4 ^b^	^A^ 51.1 ± 0.5 ^a^
**a***	0	4.26 ± 0.10 ^a^	3.63 ± 0.16 ^a^	4.04 ± 0.10 ^a^	5.13 ± 0.13 ^b^
	3	3.90 ± 0.09 ^b^	2.90 ± 0.06 ^a^	3.48 ± 0.04 ^b^	4.38 ± 0.12 ^c^
	6	3.85 ± 0.14 ^a^	3.22 ± 0.09 ^a^	3.33 ± 0.06 ^a^	4.38 ± 0.12 ^b^
	10	3.64 ± 0.07 ^a^	3.52 ± 0.10 ^a^	3.53 ± 0.09 ^a^	4.48 ± 0.22 ^b^
**b***	0	^C^ 12.7 ± 0.2	^B^ 12.6 ± 0.3	^D^ 12.6 ± 0.3	^C^ 12.9 ± 0.3
	3	^AB^ 12.1 ± 0.3 ^c^	^A^ 10.3 ± 0.3 ^a^	^C^ 11.3 ± 2.1 ^b^	^B^ 11.2 ± 0.3 ^b^
	6	^B^ 12.2 ± 0.3 ^c^	^A^ 10.4 ± 0.2 ^b^	^A^ 9.8 ± 0.1 ^a^	^A^ 10.2 ± 0.1 ^b^
	10	^A^ 11.8 ± 0.2 ^b^	^A^ 10.5 ± 0.4 ^a^	^B^ 10.6 ± 0.2 ^a^	^B^ 10.9 ± 0.4 ^a^
**MMb**	0	36.20 ± 1.83	35.54 ± 3.31	38.74 ± 1.20	38.64 ± 1.39
	3	38.84 ± 1.11	35.82 ± 2.84	40.03 ± 2.33	38.53 ± 0.63
	6	35.02 ± 0.33	36.92 ± 1.77	43.67 ± 0.30	45.65 ± 0.52
	10	37.33 ± 1.75	36.26 ± 1.60	38.67 ± 0.55	40.13 ± 0.50

^a,b,c,d^ Values within the same row with different superscripts on the right are significantly different (*p* < 0.05) between percentages of added DPP. ^A,B,C,D^ Values within the same column with different superscripts on the left are significantly different (*p* < 0.05) between storage days. DPP: date pit powder.

**Table 4 foods-09-00102-t004:** Cooking parameters of burgers with date pit powder during refrigerated storage.

	Day	Control	1.5% DPP	3% DPP	6% DPP
**Cooking yield (%)**	0	^A^ 69.9 ± 0.8 ^a^	^A^ 69.6 ± 0.9 ^a^	^A^ 76.7 ± 1.5 ^b^	^A^ 75.7 ± 1.6 ^b^
	3	^B^ 76.4 ± 1.4 ^a^	^B^ 80.1 ± 1.6 ^b^	^AB^ 80.1 ± 2.5 ^b^	^AB^ 79.7 ± 1.7 ^b^
	6	^BC^ 78.8 ± 0.6 ^a^	^B^ 80.0 ± 1.8 ^ab^	^B^ 82.9 ± 1.5 ^b^	^BC^ 83.4 ± 1.4 ^b^
	10	^C^ 80.1 ± 1.2 ^a^	^B^ 79.7 ± 1.3 ^a^	^B^ 84.8 ± 1.6 ^bc^	^C^ 86.6 ± 1.9 ^c^
**Fat retention (%)**	0	^B^ 72.3 ± 0.8 ^b^	^A^ 63.0 ± 0.8 ^a^	^A^ 80.1 ± 1.6 ^d^	^A^ 75.7 ± 1.6 ^c^
	3	^A^ 69.1 ± 1.3 ^a^	^C^ 79.3 ± 2.6 ^b^	^B^ 86.3 ± 2.7 ^c^	^B^ 87.7 ± 1.8 ^c^
	6	^C^ 77.9 ± 0.6 ^b^	^B^ 74.4 ± 1.3 ^a^	^B C^ 89.8 ± 1.7 ^d^	^B^ 85.3 ± 1.4 ^c^
	10	^D^ 83.1 ± 1.3 ^b^	^BC^ 77.2 ± 1.2 ^a^	^C^ 93.9 ± 2.9 ^c^	^B^ 83.8 ± 4.5 ^b^
**Moisture retention (%)**	0	^A^ 62.5 ± 0.7 ^a^	^A^ 64.5 ± 0.8 ^a^	^A^ 69.8 ± 1.4 ^b^	^A^ 68.0 ± 1.0 ^b^
	3	^B^ 71.1 ± 1.3	^B^ 73.4 ± 2.4	^AB^ 72.9 ± 2.3	^AB^ 71.9 ± 1.5
	6	^B^ 72.2 ± 0.5 ^a^	^B^ 74.2 ± 1.9 ^b^	^BC^ 75.8 ± 1.4 ^b^	^B^ 75.2 ± 1.5 ^b^
	10	^C^ 75.8 ± 1.2 ^b^	^B^ 72.3 ± 1.1 ^a^	^C^ 78.8 ± 1.8 ^c^	^B^ 77.2 ± 2.2 ^bc^
**Diameter reduction (%)**	0	^B^ 21.5 ± 0.5 ^c^	12.5 ± 1.6 ^a^	18.4 ± 2.7 ^bc^	^B^ 16.7 ± 2.4 ^b^
	3	^B^ 21.7 ± 0.5 ^b^	17.2 ± 2.1 ^a^	16.9 ± 2.2 ^a^	^B^ 16.9 ± 2.5 ^a^
	6	^A^ 17.1 ± 1.3 ^b^	16.2 ± 1.7 ^ab^	13.6 ± 2.9 ^a^	^AB^ 13.1 ± 2.3 ^a^
	10	^A^ 17.4 ± 1.3 ^b^	13.6 ± 2.1 ^ab^	13.3 ± 2.0 ^ab^	^A^ 10.8 ± 0.4 ^a^

^a,b,c,d^ Values within the same row with different superscripts on the right are significantly different (*p* < 0.05), between percentages of added DPP. ^A,B,C^ Values within the same column with different superscripts on the left are significantly different (*p* < 0.05) between storage days. DPP: date pit powder.

**Table 5 foods-09-00102-t005:** Sensory evaluation (trained panel) of burgers with date pit powder (DPP) during refrigerated storage.

	Day	Control	1.5% DPP	3% DPP	6% DPP
Visual color	0	^A^ 2.44 ± 0.95 ^a^	^A^ 2.51 ± 0.60 ^a^	^A^ 2.79 ± 0.81 ^a^	^B^ 3.16 ± 0.94 ^a^
	3	^B^ 3.52 ± 0.35 ^c^	^A^ 2.33 ± 0.40 ^a^	^A^ 2.66 ± 0.58 ^a^	^B^ 3.02 ± 0.67 ^ab^
	6	^C^ 4.35 ± 0.41 ^c^	^A^ 2.16 ± 0.53 ^a^	^A^ 2.41 ± 0.44 ^a^	^B^ 2.90 ± 0.41 ^b^
	10	^C^ 4.16 ± 0.80 ^d^	^B^ 3.41 ± 0.4 ^c^	^A^ 2.78 ± 0.97 ^b^	^A^ 2.33 ± 0.43 ^a^
Off-odors	0	^A^ 1.75 ± 0.54 ^a^	^A^ 1.44 ± 0.62 ^a^	^A^ 1.66 ± 0.98 ^a^	^A^ 1.44 ± 0.57 ^a^
	3	^B^ 2.45 ± 0.41 ^b^	^A^ 1.73 ± 0.81 ^a^	^A^ 1.41 ± 0.64 ^a^	^B^ 1.75± 0.88 ^a^
	6	^B^ 2.70 ± 0.44 ^b^	^A^ 1.41 ± 0.56 ^a^	^A^ 1.41 ± 0.56 ^a^	^A^ 1.48 ± 0.55 ^a^
	10	^C^ 3.61 ± 0.48 ^c^	^B^ 2.92 ± 0.34 ^b^	^B^ 2.23 ± 0.16 ^a^	^B^ 1.78 ± 0.35 ^a^

^a,b,c,d^ Values within the same row with different superscripts on the right are significantly different (*p* < 0.05), between percentages of added DPP. ^A,B,C^ Values within the same column with different superscripts on the left are significantly different (*p* < 0.05), between storage days.

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
