# Peer review of "Effect of Date (Phoenix dactylifera L.) Pits on the Shelf Life of Beef Burgers"

_foods, 2020, doi:10.3390/foods9010102_

Round 1
Reviewer 1 Report
No other comments to do
Author Response
Thank you very much for your time and considerations.
Reviewer 2 Report
Our comments on this new manuscript still as in the previous manuscript that we reviewed. Even there is no answer about our comments in the previous review: "there is no interesting and no new results in this work because the date pit powder of Phoenix dactylifera has been previously several times reported for same examinations". In this manuscript there is no discussion for similar works of previous references with present results. However, here are some remarks:
The word "Coproduct" is not suitable with the method of the work. While in the text, you use the term "date pit powder DPP" Some typos mistakes are found example: Line 339 and line 430: Table 4 ; correct the abbreviation "PDP" Line 430: the number of the Table is repeated Some data in the tables are not clear; e.g. Line 430: what means "1.5% DPP", if units of weight were not given in the method Line 430: What meant the results (e.g. 2.44 ± 0.95) of visual color and off-odor; no units and no discuss for these results. Some data given in other meaning, e.g. Table 1: "chemical composition" title is not suitable for "moisture and ash"Author Response
Dear reviewer 2:
We thank for your comments. We have rewritten the text, the title and the references to incorporate your recommendations and the suggestions.
Review: Our comments on this new manuscript still as in the previous manuscript that we reviewed. Even there is no answer about our comments in the previous review: "there is no interesting and no new results in this work because the date pit powder of Phoenix dactylifera has been previously several times reported for same examinations". In this manuscript there is no discussion for similar works of previous references with present results. However, here are some remarks:
Authors: In the introduction we have written a new sentence justifies the novelty of this study. We have rewritten in the discussion referring to similar works All these changes are highlighted in the text in green color.
Review: Coproduct is not suitable with the method of the work. While in the text, you use the term "date pit powder DPP"Authors We have rewritten the title “Effect of date (Phoenix dactylifera L.) pits on the shelf-life of beef burgers” and in the abstract “date palm coproducts (pits)” is in the text in green color
Review: Line 339 and line 430: Table 4 ; correct the abbreviation "PDP"
Authors: We apologize for the mistake. Now corrected.
-Review: Line 430: the number of the Table is repeated
Authors We apologize for the mistake. Now corrected.
Review: Line 430: what means "1.5% DPP", if units of weight were not given in the method Line 430: What meant the results (e.g. 2.44 ± 0.95) of visual color and off-odor; no units and no discuss for these results.
Authors: In the sensory analysis we used a scale from 0 to 7 for the untrained panelists and from 1 to 5 for trained panel. Now there are discussions for these results.
Review: Some data given in other meaning, e.g. Table 1: "chemical composition" title is not suitable for "moisture and ash"Authors: We think that the most relevant results of the DPP characterization refer to the fiber content and the antioxidant capacity of date pit, hence the highlight of section 3.1.
Thank you very much for your time and considerations.
Reviewer 3 Report
I think this manuscript is acceptable in this present form.
Author Response
Thank you very much for your time and considerations.
This manuscript is a resubmission of an earlier submission. The following is a list of the peer review reports and author responses from that submission.
Round 1
Reviewer 1 Report
i suggest you to better describe the method regarding the Pigment oxidation (Metmyoglobin percentage). A reader cannot go to find the original reference, but should be capable to repeat your experiment simply reading your work.
For the other, I don't suggest anything
Reviewer 2 Report
The author worked with the "Effect of date (Phoenix dactylifera L.) pit powder on safety and quality of beef burger". However, the seed of Phoenix dactylifera was previously reported about its effect to persevere meat, about its antioxidants, phenolic contents and its nutritional values. Therefore, there are no new interesting data in this work about the plant seeds; even the work is written well.
see previous reports:
Essa, R.Y. and Elsebaie, E., 2018. Effect of using date pits powder as a fat replacer and anti-oxidative agent on beef burger quality. J Food and Dairy Sci, Mansoura Univ, 9, pp.91-6. Amany, M.M. B., Shaker,M.A., & Abeer, A. K. (2012). Antioxidant activities of date pits in amodel meat system. International Food Research Journal, 19, 223–227. Sadiq, I.S., Izuagie, T., Shuaibu, M., Dogoyaro, A.I., Garba, A. and Abubakar, S., 2013. The nutritional evaluation and medicinal value of date palm (Phoenix dactylifera). Int J Mod Chem, 4, p.147e154. Habib, H.M., Platat, C., Meudec, E., Cheynier, V. and Ibrahim, W.H., 2014. Polyphenolic compounds in date fruit seed (Phoenix dactylifera): characterisation and quantification by using UPLC‐DAD‐ESI‐MS. Journal of the Science of Food and Agriculture, 94(6), pp.1084-1089.Reviewer 3 Report
The study aims to determine the effects of incorporating date pits by-product on the physicochemical, microbial and sensory properties of beef burgers. The results of the paper are interesting and may provide a basis for use of novel plant-based ingredients in meat products in the future. However, some revisions should be made.
Specific comments
P1, L21: Please change 1% to 1.5% P1, L41-42: Not clear on the relationship between meat preparation and higher risk of oxidation. The author may need to explain the theoretical details on why emulsified meats are more prone to lipid oxidation and microbial contamination. P2, L73: Suggest the authors to define date pits and what differentiates it from the date seed discussed in the previous paragraph. P3, L104: Suggest to use g to allow repetition in future studies. Please revise for the rest of centrifugation-related procedure. P3, L111: Please check if the ‘buffered egg yolk’ part is necessary. Also suggest to provide the full name of TBARS. P3, L119: The formulation doesn’t add up to 100%. Moreover, what is the proportion (not concentration) of DPP added into the burger? Using a table to list the composition of each treatment may be helpful for readers to understand. P5, L191-197: Suggest to specify that these panellists are ‘untrained’. Please provide the number (1-7) for the hedonic scale, and specify which number is for highly like/dislike. This may avoid the confusion with the scale used for visual/off-odor analysis. Table 1: Please check the values for insoluble dietary fiber. P6, 218-220: What constitutes as a ‘high’ value? And how does it compare to previous studies? P7, L237: Please change ‘in’ to ‘by’. Table 2: Please add superscripts for MMB of cooked samples, if there’s any statistical difference. Why does the moisture content for the cooked samples increased at 1.5% DPP, but showed a decreasing trend at 3 and 6% DPP? Additionally, cooking results in opposite trends in fat content of different treatments (i.e. 1.5% DPP vs the rest), despite the fat drip loss. The authors may need to explain this unexpected finding. P8, L263-265: Specify if it is for the cooked samples. Aside for the 1.5% DPP, there is no significant difference in L values between raw control and DPP-added burgers (there’s even an increase at the 3rd day of storage). There’s also some fluctuations in L values during the storage period. The authors may need to explain more. P10, L313-314: Sentence may be simplified to ‘These lower values could be related to the date pit fibre’s retention of water and fat in the meat matrix’. P10, L317: Remove ‘when cooked at home’. P10, L326: Remove ‘as you would expect’. P12, L345: Please insert a space for ‘wereprobably’ Figure 2: Suggest to use ‘.’ instead of ‘,’ for decimal place P13-14, L379-390: Suggest to make a table for color and off-odor analysis to make it clearer to readers. P14, L397: Please briefly list the possible underlying reasons for the lower acceptance scores of 6% DPP (i.e. texture, etc.). P14, L400-401: Not clear on ‘but they had a lower L* that lower concentrations’
Reviewer 4 Report
This is nicely performed. I would recommend this accepted as is.